# Clinical Comparative Study of Shade Measurement Using Two Methods: Dental Guides and Spectrophotometry

**DOI:** 10.3390/biomedicines12040825

**Published:** 2024-04-09

**Authors:** Alfonso Alvarado-Lorenzo, Laura Criado-Pérez, Mónica Cano-Rosás, Eva Lozano-García, Juan López-Palafox, Mario Alvarado-Lorenzo

**Affiliations:** 1Department of Oral Surgery, Universidad de Salamanca, 37007 Salamanca, Spain; lauracriado@usal.es (L.C.-P.); mcanorosas@usal.es (M.C.-R.); 2Department of Health Sciences, Miguel de Cervantes European University of Valladolid, 47012 Valladolid, Spain; elozano@uemc.es; 3Department of Dentistry, Universidad Alfonso X el Sabio, 28691 Villanueva de la Cañada, Spain; jlopepal@uax.es; 4Department of Dentistry, Universidad Católica San Antonio de Murcia, 30107 Murcia, Spain; malvarado@pgoucam.com

**Keywords:** color vision, shade matching, spectrophotometer, visual shade match, shade guide

## Abstract

Tooth color is a determining factor in the fabrication of dental prostheses. The aim of the present study is to compare two measurement methods used in the field of dentistry: dental guides and spectrophotometry. A total of 2768 natural teeth were measured using the Vita Classical and Vita 3D-Master dental guides (Vita-Zahnfabrik, Bad Säckingen, Germany), as well as a Vita Easyshade Compact spectrophotometer (Vita-Zahnfabrik). The measurements were carried out by one operator under suitable illumination conditions at 5500 degrees Kelvin. The obtained results show that the measurements obtained with the Vita Classical dental guide classifies teeth into the A-B categories, while the spectrophotometer preferentially classifies teeth into the B-C categories. The correlation coefficients obtained with the dental guides ranged from −0.32 to −0.39 (*p* < 0.01), while those for the spectrophotometer ranged from −0.35 to −0.55 (*p* < 0.01). Therefore, we can conclude that the spectrophotometer is more reliable and reproducible in its measurements than the dental guides.

## 1. Introduction

Tooth color is part of people’s physical attractiveness, as smiling is the first approach in non-verbal communication. Therefore, in order to successfully meet this challenge—even more so at a time when our profession is required not only to rehabilitate function but also to provide the best possible aesthetics—the dentist must apply the science of color, especially colorimetry, the branch of color science that deals with the methods and techniques for measuring color, evaluating light energy in terms of the sensation of color that it produces in the human eye [1,2].

In dentistry, two types of colorimetric methods are used: one uses dental guides for subjective color measurement, while the other uses measuring devices, spectrophotometers, and colorimeters for objective color measurement [3,4,5,6,7]. The most widely applied method in clinical dentistry for color communication during the fabrication of indirect restorations is visual color determination through subjective color measurement [5,8,9]. The shades of a commercially available color guide are compared to the tooth adjacent to the one that needs restoration. The color that most closely resembles the tooth is then communicated to the dental laboratory using verbal, graphic, and photographic means [10].

The knowledge and management of color science are thus essential for the professional who wants to perform modern, aesthetic, and functional dentistry at the same time. To this day, the subjective colorimetric technique—when used correctly—is a sine qua non for the success of restorations. However, the limitations of this technique are evident, such as influencing factors (e.g., room lighting), visual phenomena, tooth morphology, light reflection, and operator fatigue [7,9,10,11,12,13,14].

The use of digital color measurement in dentistry seems to be a good proposal for the objective measurement of color, as has been demonstrated in other sciences and industrially [15]. However, in dentistry, its adoption seems to be slower due to the mistrust and cost of the methods used [16,17]. Dr. E. Bruce Clark was the first to subject natural teeth to the measurement and scientific analysis of color, communicating the importance of color dimensions when he stated in 1931 that, in the study of color, the consideration of its three dimensions—lightness or value, chroma, and hue—is not only a basic requirement but also the most important one [13]. The use of a spectrophotometer on vital teeth, comparing them with extracted teeth, was proposed by Loyd Miller in 1984. A spectrophotometer measures the intensity of the wavelength visible to the human eye by emitting a standard light and then interpreting it into the C.I.E. coordinates of the International Color Commission. Therefore, we can consider these devices as simulators of the human eye, thus increasing the reliability of their results [18].

According to Dr. G. Henning—a graduate in physics from the Dental Engineering Base—a study involving 162 clinicians and 178 laboratory technicians found that 58% of color definitions were incorrect. Digital color analysis is intended to help replace the subjective sensation of the human eye, providing exactly reproducible data for the construction and manufacture of dental prostheses. In the devices currently available on the market, different measurement principles are applied, which involve emitting light and measuring its reflection [16,19,20,21].

Several studies have also mentioned the importance of shade changes caused by caries and trauma [22]. However, there are also different restorative materials in implant prosthodontics that must be considered in order to create natural and functional oral restorations [23]. All of these factors must be taken into account by restorative dentists, who must fabricate prostheses with an understanding of the variability that they can undergo in the oral environment due to the bacterial flora [22].

The reliability of the various restorative systems is a challenge, and, at present, the aesthetic demands of patients are very high [24,25]. Therefore, shade selection can now be performed with the use of dental guides, spectrophotometers, and digital intraoral scanners, with the latter being able to capture and differentiate between hard and soft tissue [26,27,28,29]. However, despite technological advances, there are several studies that have spoken of the importance of verifying the objectivity of color acquisition by means of measuring devices, in contrast to subjective colorimetry, confirming that the selected color is the right one and, thus, enabling accurate communication with the laboratory that manufactures the prostheses [30,31].

The aim of our study was to evaluate the concordance of the two methods for color measurement used in dentistry—using dental guides (Vita Classical and Vita Toothguide 3D-Master, Vita Zahnfabrik) and spectrophotometry (Vita Easyshade Compact, Vita-Zahnfabrik)—in relation to the lightness or value of natural teeth. The null hypothesis of the study is that there is no difference in measuring color using dental guides and spectrophotometry.

## 2. Materials and Methods

### 2.1. Study Design

The study followed the ethical standards of the Declaration of Helsinki for biomedical research and was approved by the Ethics Committee for Research with Medicines in the Health Area of Valladolid (reference: PI 20-1911 NO HCUV Ethics Committee of the Clinical University Hospital of Valladolid).

A cross-sectional clinical study was conducted, in which 2768 natural teeth were measured in 294 patients—123 men (41.8%) and 171 female (58.2%)—with an average age of 34.48 years (3.6 SD). Color was measured objectively using the Vita Easyshade Compact spectrophotometer (Vita-Zahnfabrik, Bad Säckingen, Germany) and subjectively with the Vita Classical A1-D4 and Vita Toothguide 3D-Master (Vita-Zahnfabrik, Bad Säckingen, Germany) color guides. Previously published studies were considered when determining sample size [32,33,34].

### 2.2. Patient Selection

Of the 294 Spanish Caucasian patients, the inclusion criteria of the study subjects were patients with permanent dentition preferably, with the following natural teeth: left and right upper central incisor, right and left upper lateral incisor, right and left upper canine, right and left upper first premolar and right and left upper second premolar. The exclusion criteria were: teeth with restorations, veneers or crowns, endodontic teeth, teeth with orthodontic retention on the palatal face, whitened teeth or teeth with inability to take color, patients taking medication that may alter tooth color, and heavy smokers and/or coffee drinkers who have stained enamel surfaces.

### 2.3. Measurement Process

The color measurement was performed by an observer with 10 years of experience and instructed in the science of color in dentistry, without deficiencies in color vision, for which the Ishihara test was previously applied [35]. It was carried out in a 10 m^2^ cabinet, illuminated with artificial light with white light and natural light through two windows (1 m wide by 1.5 m high). Measurements were made inside the cabinet using a Sekonic dual spot l-778 photometer (Sekonic Co., Tokyo, Japan) to locate the areas of approximately 5500 degrees Kelvin.

In the subjective color measurement process, the color was selected using Vita Classical Vita 3D-Master guides, while objective color measurement was carried out using the Vita Easyshade Compact spectrophotometer. With the tooth hydrated, the tip of the spectrophotometer should be perpendicular to the vestibular aspect of the tooth analyzed, and it should not be moved until the results of the color analysis have been obtained. In the subjective and objective colorimetry measurements, three measurements were made on each tooth by each observer. The waiting period between sampling was 15 min. Measurements were taken with the dental guides in areas of the room that avoided reflections and shadows. Values that matched the three measurements were recorded on the data collection sheet and each selected tooth was measured using both methods. In order to analyze the data in terms of lightness or value of the guide, Vita Classical was ordered into ordinal values as follows [36]: B1 (15), A1 (14), A2 (13), D2 (12), B2 (11), C1 (10), C2 (9), D4 (8), D3 (7), A3 (6), B3 (5), A3.5 (4), B4 (3), C3 (2), A4 (1).

### 2.4. Statistical Analysis

Regarding the statistical analysis, the parameters obtained in the research were entered into a database—in this case, using the Microsoft 365 (Excel 16) computer program for Windows 10—and the statistical analysis was carried out with the SPSS 28 program. To obtain the descriptive data, percentages per tooth and clarity averages were calculated. Pearson’s correlation coefficient was used to compare human judgments and spectrophotometers at a statistically significant level of *p* < 0.01.

## 3. Results

Table 1 shows the results of the lightness measurement obtained with the Vita Classical dental guide. We can see that the tooth with the highest percentage of frequency was the color B3 for the second lower premolar, and the lowest was the upper and lower central incisor with 0%. Furthermore, we can analyze from the measurements that the anterior teeth were selected in the A1–A2 and B1–B2 categories, while the posterior teeth were preferentially selected in the A3–A3.5 and B3–B4 categories. In general, the least selected frequencies were in category D.

Table 2 shows the results of the measurement of the value or lightness of the dental guide using a spectrophotometer. We can see that the tooth with the highest percentage of frequency was the color B3 for the lower canine, and the lowest was all teeth in one of the groups with 0%. Furthermore, from the table, we can analyze that the anterior teeth were selected in the A1–B2 and C1–C2 categories when measured using the spectrophotometer, while the posterior teeth were selected in the B3–B4 categories, with the least selected frequencies being generally those in the D category (in this case, coinciding with the humans and the measurement with the dental guides).

Table 3 shows the total comparative percentages between dental or human guides and spectrophotometry. The difference is that, while humans most frequently select categories A and B, the spectrophotometer selected categories B and C, with 53% of the teeth being from B only. Group D had the lowest frequency in the selection of both groups.

Table 4 shows the mean of the value or lightness obtained with the Vita 3D Master compared to the spectrophotometer. There were five lightness groups, with group 1 being the lightest and group 5 the darkest. The anterior teeth were lighter than the posterior teeth in both groups, with the upper central incisor being selected with the highest lightness (2.194). The darkest was the second upper premolar, with a mean of 3.202. We can see that there were differences between the two methods, according the highest values obtained for the spectrophotometer measurements; therefore, they were darker than those selected using the dental guides (also shown in Figure 1 by category).

Table 5 shows the differences between the two measurement methods. The correlation coefficients in the dental guide method were highest for the lower central incisor, at −0.39 (*p* < 0.01), while −0.55 (*p* < 0.01) was obtained for the upper central incisor in the spectrophotometer. The lowest value with the guide was for the upper first premolar, at −0.29 (*p* < 0.01), and the lower first premolar, at −0.39 (*p* < 0.01) for the spectrophotometer. The correlations were higher for the measurements made with the spectrophotometer than those with the dental guides; therefore, it can be considered more reliable in its measurements (also shown in Figure 2).

## 4. Discussion

The obtained results show that there are differences in measurement between the two methods. In particular, the dental guides had a lower correlation in their measurements compared to the spectrophotometer; therefore, we can consider that the objective measurement is more reliable and reproducible than the subjective one. Many studies have shown that measurement with subjective methods is less reliable and accurate than when using measuring devices [10,12,37,38,39,40,41]. Dental guides have also been shown to be inconsistent with the color spectrum of natural teeth [13,42,43,44].

In the study by Hasegawa, Akira et al. in 2000 [14], using a colorimeter and comparing it with the Vita Lumin Vacuum guide in a sample of 87 patients, he concluded that the distribution of natural teeth does not correspond to the Vita guide. The same results were reached by Guo H. et al. [45], with a sample of 15,836 restorations and 138 observers using the Vita guide. The problem is that the Vita guide was created based on samples from Central European teeth and, so, there will be an obvious discrepancy with the teeth of Eastern patients. A study conducted at Kuwait University concluded that there is no difference between the measurement of color by dental students and trained and experienced professionals [46].

The reliability of measuring devices has been demonstrated in previous studies. The first was conducted by Tung, Francis F. et al. in 2002 [29]. In the first part of their study, two examiners used the Shade Eye-Ex Chroma Meter (Shofu Inc., Kyoto, Japan) to measure the upper right centers of 11 people. Three weeks later, the measurements were taken again. In the second part of the study, the same was carried out with two experienced observers. A total of 82% of the measurements conducted with the colorimeter were reproducible, while observers agreed on 73%. In another study, conducted this year by Paul, S. et al. [10], the sample consisted of 30 patients and the natural upper incisors without restorations were measured. In the group of observers, the coincidence was 26.6% of the measurements; however, with the spectrophotometer the coincidence was 83.3% of the measurements.

Lightness is the first dimension that the human eye sees. It is important to note that, in the comparison of the classical guide with the spectrophotometer, humans classify teeth more clearly than the spectrophotometer in the classic guide; the same is true for the comparison with the 3D Master guide. Brightness or luminosity is the one of the three dimensions of color that the human eye can perceive with the greatest concordance in subjective color perception with dental guides, followed by color and, finally, intensity [47].

However, the results obtained are similar to other previously published studies concluding that spectrophotometers and the 3D-Master guide obtain optimal results for use in dental shade-taking, when compared to other dental guides [48]. In the present study, the lowest value of the Kappa index in reproducibility was obtained for the Vita Classical guide, with a value of 0.177, while the spectrophotometer had a value of 0.805. In our study, the correlation according to the Pearson index was in the range from −0.36 to −0.39 for humans and −0.35 to −0.55 for the spectrophotometer. Therefore, the measurements obtained using measuring devices are more reproducible than when using dental guides, which is logical; however, the values obtained indicate that both the dental guides and the measuring devices have moderate values clinically, indicating that they should be improved in order to make prostheses that are as similar as possible to their adjacent natural teeth, especially in the former sectors where the aesthetic requirements are higher (as concluded in other articles [49]). We must, therefore, use measuring devices such as spectrophotometers to obtain greater reliability and reproducibility in our clinical measurements.

The latest developments are intraoral scanners, which help to take images of the teeth and send them to the laboratory for better communication, avoiding the need for plaster models [50]. There are also studies on the application of intraoral scanners for shade selection, comparing it with colorimeters and spectrophotometers [11,51]. In the results, digital methods, both scanners and spectrophotometers are reliable in color measurement in dentistry, however, they must be confirmed by the visual method [11,29,31,51].

Regarding the limitations of the study, we consider that the measurements could be made with different spectrophotometers and colorimeters (and even with dental scanners) in order to help us conduct color mapping of the teeth [29]. The results could be seen as evidence that the use of technology helps to improve measurements in cases where the human eye cannot differentiate. However, the correlations obtained with the objective measurement were still less than 0.60 for all teeth, which we consider be low; therefore, the measuring devices should also be improved. To the best of our knowledge, no studies have been conducted with the sample size used in our study, as well as analyzing anterior and posterior teeth in both dental arches.

## 5. Conclusions

From the results of the present study, it can be concluded that there are differences between the two methods of measurement. When measuring with dental guides, people tend to classify natural teeth into categories A and B, whereas the spectrophotometer prefers to classify them into categories B and C. Overall, measuring teeth using a spectrophotometer results in a lower whiteness value and the measurements are more reproducible than those obtained with dental guides.

## Figures and Tables

**Figure 1 biomedicines-12-00825-f001:**
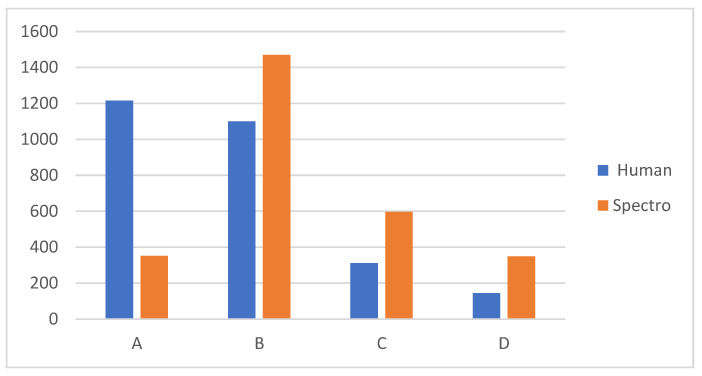
The total number of teeth measured, divided into categories of color assessed using dental guides (Vita Classical and spectrophotometer).

**Figure 2 biomedicines-12-00825-f002:**
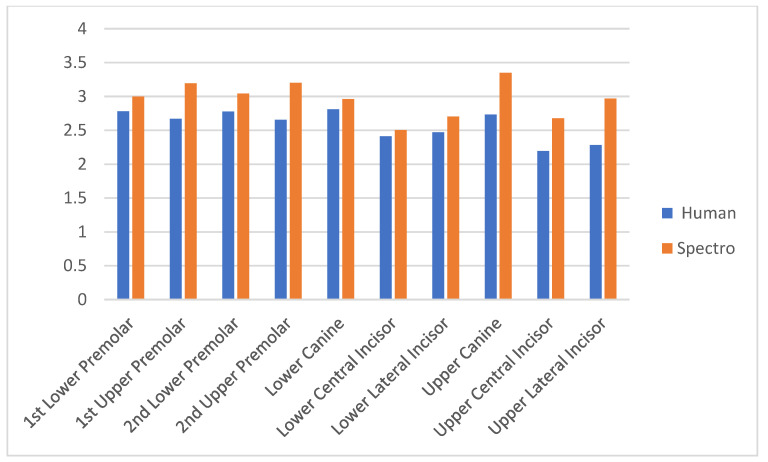
The average of each type of tooth, in terms of the lightness measured with the 3D vita guide and spectrophotometer.

**Table 1 biomedicines-12-00825-t001:** Descriptive data (in percentage by row) for each of the teeth assessed with dental guides (Vita Classical).

Type of Tooth	A1	A2	A3	A3.5	A4	B1	B2	B3	B4	C1	C2	C3	C4	D1	D2	D3	D4
Upper Central Incisor	16%	13%	6%	2%	0%	27%	15%	1%	0%	13%	2%	0%	0%	0%	3%	2%	0%
Lower Central Incisor	8%	23%	10%	4%	1%	11%	19%	4%	0%	8%	4%	2%	1%	0%	4%	1%	0%
Upper Lateral Incisor	8%	16%	6%	1%	0%	14%	26%	4%	1%	13%	5%	1%	0%	0%	3%	1%	0%
Lower Lateral Incisor	5%	19%	14%	5%	2%	7%	22%	8%	1%	4%	7%	2%	1%	0%	2%	1%	0%
Upper Canine	0%	12%	18%	16%	2%	1%	11%	22%	5%	0%	5%	2%	0%	0%	1%	2%	1%
Lower Canine	0%	10%	15%	17%	7%	0%	8%	22%	12%	0%	4%	1%	0%	0%	1%	1%	1%
1st Upper Premolar	2%	13%	15%	11%	0%	2%	20%	13%	3%	3%	6%	1%	0%	0%	5%	5%	0%
1st Lower Premolar	1%	9%	23%	16%	2%	0%	11%	22%	6%	1%	4%	1%	0%	0%	1%	3%	1%
2nd Upper Premolar	3%	13%	16%	8%	1%	3%	21%	12%	1%	4%	7%	2%	1%	0%	3%	4%	1%
2nd Lower Premolar	1%	10%	24%	12%	3%	1%	10%	24%	6%	0%	6%	1%	0%	0%	1%	1%	1%

**Table 2 biomedicines-12-00825-t002:** Data (in percentage by row) for each of the teeth measured using the spectrophotometer.

Type of Tooth	A1	A2	A3	A3.5	A4	B1	B2	B3	B4	C1	C2	C3	C4	D1	D2	D3	D4
Upper Central Incisor	13%	1%	0%	0%	1%	2%	18%	1%	1%	10%	14%	7%	0%	0%	29%	0%	1%
Lower Central Incisor	18%	2%	0%	0%	0%	3%	19%	2%	0%	12%	9%	1%	0%	0%	31%	0%	1%
Upper Lateral Incisor	3%	1%	1%	0%	1%	1%	16%	7%	2%	14%	23%	12%	0%	0%	15%	0%	4%
Lower Lateral Incisor	5%	1%	1%	0%	0%	1%	33%	10%	3%	14%	15%	2%	0%	0%	12%	0%	2%
Upper Canine	1%	0%	1%	5%	13%	0%	6%	22%	31%	3%	5%	8%	1%	0%	1%	0%	3%
Lower Canine	0%	1%	1%	1%	4%	0%	11%	41%	31%	1%	4%	2%	0%	0%	0%	0%	1%
1st Upper Premolar	0%	1%	2%	5%	5%	0%	7%	32%	23%	1%	7%	10%	1%	0%	2%	0%	5%
1st Lower Premolar	0%	1%	2%	3%	3%	0%	11%	36%	31%	1%	4%	4%	0%	0%	0%	0%	4%
2nd Upper Premolar	0%	0%	3%	4%	9%	0%	7%	28%	25%	0%	7%	9%	1%	0%	0%	0%	6%
2nd Lower Premolar	0%	0%	3%	3%	4%	0%	10%	34%	29%	1%	3%	8%	0%	0%	0%	1%	3%

**Table 3 biomedicines-12-00825-t003:** Percentages of the total number of teeth measured, divided into categories of color assessed using dental guides (Vita Classical and spectrophotometer).

Category	Human	Spectrophotometer
A	1215 (44%)	352 (13%)
B	1099 (40%)	1470 (53%)
C	311 (11%)	597 (22%)
D	144 (5%)	349 (13%)

**Table 4 biomedicines-12-00825-t004:** The average of each type of tooth, in terms of the lightness measured according to the 3D vita guide and spectrophotometer.

Type of Tooth	Human (Mean-SD)	Spectrophotometer (Mean-SD)
Upper Central Incisor	2.194 (0.63)	2.677 (0.59)
Lower Central Incisor	2.413 (0.72)	2.505 (0.56)
Upper Lateral Incisor	2.284 (0.67)	2.968 (0.54)
Lower Lateral Incisor	2.471 (0.70)	2.702 (0.53)
Upper Canine	2.733 (0.68)	3.350 (0.54)
Lower Canine	2.811 (0.73)	2.963 (0.52)
1st Upper Premolar	2.670 (0.72)	3.195 (0.53)
1st Lower Premolar	2.782 (0.72)	2.998 (0.50)
2nd Upper Premolar	2.655 (0.70)	3.202 (0.52)
2nd Lower Premolar	2.777 (0.70)	3.042 (0.50)

**Table 5 biomedicines-12-00825-t005:** The correlations for each type of tooth with the luminosity measured using the spectrophotometer vs. measurement with dental guides (Vitapan 3D and Classical).

Type of Tooth	Human	Spectrophotometer
Upper Central Incisor	−0.34 **	−0.55 **
Lower Central Incisor	−0.39 **	−0.48 **
Upper Lateral Incisor	−0.37 **	−0.43 **
Lower Lateral Incisor	−0.39 **	−0.35 **
Upper Canine	−0.29 **	−0.48 **
Lower Canine	−0.38 **	−0.45 **
1st Upper Premolar	−0.29 **	−0.40 **
1st Lower Premolar	−0.35 **	−0.39 **
2nd Upper Premolar	−0.26 **	−0.40 **
2nd Lower Premolar	−0.32 **	−0.39 **

** Statistically significant results (*p* < 0.01).

## Data Availability

Data are only available on request due to restrictions, e.g., privacy or ethical restrictions.

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
