# Peer review of "Clinical Comparative Study of Shade Measurement Using Two Methods: Dental Guides and Spectrophotometry"

_biomedicines, 2024, doi:10.3390/biomedicines12040825_

Round 1

Reviewer 1 Report

Comments and Suggestions for Authors

Work of sufficient scientific value on different tooth color detection systems. Some important criticisms are listed below:

-Check that all keywords are Pubmed mESH terms

-At the end of the introduction section, the null hypotheses of the study must be inserted which will be refuted in light of the results obtained

-The introduction section is extremely simple. Some important considerations are missing both on color variability in endodontically treated elements and, for example, in color detection in the case of dental implants. In this regard, I recommend including the following works in the reference section which may be of help to the reader:

Grande F, Pozzan MC, Marconato R, Mollica F, Catapano S. Evaluation of Load Distribution in a Mandibular Model with Four Implants Depending on the Number of Prosthetic Screws Used for OT-Bridge System: A Finite Element Analysis (FEA). Materials (Basel). 2022 Nov 10;15(22):7963. doi: 10.3390/ma15227963. PMID: 36431449; PMCID: PMC9699052.

Valenti C, Pagano S, Bozza S, Ciurnella E, Lomurno G, Capobianco B, Coniglio M, Cianetti S, Marinucci L. Use of the Er:YAG Laser in Conservative Dentistry: Evaluation of the Microbial Population in Carious Lesions. Materials (Basel). 2021 May 4;14(9):2387. doi: 10.3390/ma14092387. PMID: 34064339; PMCID: PMC8124663.

These problems make a work that is otherwise too simplistically focused on a single chromatic aspect more interesting. In this regard, it would be interesting for the authors to add what is said in the scientific literature about these two specific situations.

-The part online 80 on the ethics committee must be moved to the beginning of the materials and methods

-Some additional considerations should also be added on the problem relating to obtaining tooth color in the presence of facial scanners; digital technology, in fact, has represented an important evolution but also a great risk in this regard. The authors must also examine this aspect.

-In general, a general revision of the English language should be carried out, certifying the process.

Comments on the Quality of English Language

extensive english check must be added

Author Response

Work of sufficient scientific value on different tooth color detection systems. Some important criticisms are listed below:

- Question: Check that all keywords are Pubmed mESH terms

- Answer: The keywords have been revised, thank you for your comment.

- Question: At the end of the introduction section, the null hypotheses of the study must be inserted which will be refuted in light of the results obtained

- Answer: We have added the null hypothesis of the paper at the end of the introduction, thank you for the suggestion.

- Question: The introduction section is extremely simple. Some important considerations are missing both on color variability in endodontically treated elements and, for example, in color detection in the case of dental implants. In this regard, I recommend including the following works in the reference section which may be of help to the reader:

Grande F, Pozzan MC, Marconato R, Mollica F, Catapano S. Evaluation of Load Distribution in a Mandibular Model with Four Implants Depending on the Number of Prosthetic Screws Used for OT-Bridge System: A Finite Element Analysis (FEA). Materials (Basel). 2022 Nov 10;15(22):7963. doi: 10.3390/ma15227963. PMID: 36431449; PMCID: PMC9699052.

Valenti C, Pagano S, Bozza S, Ciurnella E, Lomurno G, Capobianco B, Coniglio M, Cianetti S, Marinucci L. Use of the Er:YAG Laser in Conservative Dentistry: Evaluation of the Microbial Population in Carious Lesions. Materials (Basel). 2021 May 4;14(9):2387. doi: 10.3390/ma14092387. PMID: 34064339; PMCID: PMC8124663.

These problems make a work that is otherwise too simplistically focused on a single chromatic aspect more interesting. In this regard, it would be interesting for the authors to add what is said in the scientific literature about these two specific situations.

- Answer: These chromatic change factors have been added in the introduction on lines 70-84 and new, more up-to-date references have been incorporated. Thank you for the suggestion.

 -The part online 80 on the ethics committee must be moved to the beginning of the materials and methods

- Answer: The reviewer's proposal has been carried out and the paragraph on the ethics committee at the beginning of the material and methods section has been changed.

- Question: Some additional considerations should also be added on the problem relating to obtaining tooth color in the presence of facial scanners; digital technology, in fact, has represented an important evolution but also a great risk in this regard. The authors must also examine this aspect.

Answer: We have added a paragraph to the discussion on lines 243-248 explaining the current developments of scanners compared to spectrophotometry.

-In general, a general revision of the English language should be carried out, certifying the process.

-We have conducted a review with mpdi's English language editors.

Thank you

Reviewer 2 Report

Comments and Suggestions for Authors

In this study, the two methods of color measurement to the lightness of natural teeth was reported. Some issues need to be addressed.

Materials and Methods

1. The total number of teeth is 2768. What are the number of specimens for each type of tooth (for each group)?

2. For the exclusion criteria, did you exclude heavy smokers and heavy coffee drinkers?

3. For statistical analysis, what is the significant level of p?

Results

1. In Table 4, please add the significant symbols to indicate if there are any significant differences between groups. Please also check it is mean or media reported in the table. No mean ± SD reported in the table?

Discussion

1. p. 6 of 8, line 181, ref 36 reported that there is no difference between the measurement of colour by dental students and trained and experienced professionals. Did you find other studies that there is actually difference between experienced and non-experienced professionals?  The outcome of colour measurement may be affected by the years of experience of observer. For example, 10 years vs. 25 years.  

2. From Table 5, the correlations for all groups were between -0.3 and -0.5. That means there is a low correlation using dental guides. Is it still commonly used method today? Any advantages over spectrophotometry if they are still commonly used.

Other comments

1. English editing is recommended, e.g., p.6 of 8, line 214, “The results found could be found evident as …..”.

Comments on the Quality of English Language

1. English editing is recommended.

Author Response

- In this study, the two methods of color measurement to the lightness of natural teeth was reported. Some issues need to be addressed.

- Materials and Methods

-  Question: The total number of teeth is 2768. What are the number of specimens for each type of tooth (for each group)?

  • Answer: A cross-sectional clinical study was conducted in which 2768 natural teeth were measured in 294 patients. The following table shows the number of teeth grouped with the classification of the vita classical guide. However, we thought it more convenient to put the percentages in the article. In both groups the sample size was the same, each tooth was measured by dental guides and spectrophotometry. A comment has been incorporated in line 127-130.
  • - Question: For the exclusion criteria, did you exclude heavy smokers and heavy coffee drinkers?

    - Answer: Yes, patients who drank tea or coffee in high doses and smokers were excluded. However, all patients underwent an oral cleaning prior to shade taking to remove stains that were derived from certain habits and could interfere with tooth shade taking. We have attached in material and methods on line 111-112 or clarification.

    - Question: For statistical analysis, what is the significant level of p?

    - Answer: Statistically significant level was p < 0.01. We have attached in material and methods on line 138-140 for clarification.

    Results

    - Question:  In Table 4, please add the significant symbols to indicate if there are any significant differences between groups. Please also check it is mean or media reported in the table. No mean ± SD reported in the table?

    - Answer: The standard deviation has been added to the table with the descriptive data. Thank you for your appreciation.

    Discussion

    - Question: p. 6 of 8, line 181, ref 36 reported that there is no difference between the measurement of colour by dental students and trained and experienced professionals. Did you find other studies that there is actually difference between experienced and non-experienced professionals?  The outcome of colour measurement may be affected by the years of experience of observer. For example, 10 years vs. 25 years.  

    - Answer: Dear reviewer, there are many studies on this subject. The years of experience and the gender of the sample are not determining factors in colour measurement, as in professionals with more than 25 years of experience, the vision can be altered by the years of work and negatively influence the colour measurement. We recommend the following article for further information.

    Aswini KK, Ramanarayanan V, Rejithan A, Sajeev R, Suresh R. The effect of gender and clinical experience on shade perception. J Esthet Restor Dent. 2019 Nov;31(6):608-612. doi: 10.1111/jerd.12520. Epub 2019 Aug 27. PMID: 31456329.

    -Question: From Table 5, the correlations for all groups were between -0.3 and -0.5. That means there is a low correlation using dental guides. Is it still commonly used method today? Any advantages over spectrophotometry if they are still commonly used.

    Answer: In most of the articles found for more than 20 years, the correlation has been low using dental guides. Spectrophotometry is a real alternative with better results. However, the cost of measuring devices and the work of laboratory technicians makes the tooth colour in restorations acceptable in most cases. But if we are looking to make prostheses that are not detectable to the human eye, we should use intraoral scanners and measuring devices that are more commonly used nowadays.

    Other comments

  • English editing is recommended, e.g., p.6 of 8, line 214, “The results found could be found evident as …..”
  • An English language review has been carried out with mdpi.
  • Thank you

Reviewer 3 Report

Comments and Suggestions for Authors

Tooth colour is an important  factor in the fabrication of dental prostheses, despite the fact that may be is not  a determining one as afirmed  the paper abstract. But of course is an interesting subject for both type of target groups uch as patients and  people working in the field. This is a paper merit, The goal  of the present study is evaluation   of the two methods of color measurement in dentistry, using dental guides and spectrophotometry respectively. Such  proposed research design  is also a good idea but the way to support and to performed the investigation need to be more rich, Before publication the paper has to be completed according to the following

a) introduction is based on the old references the majority of titles being older than 15 years.  

b) methods have to be performed in a modified research design. Selection of patients it  will be better to be not larger but more complex, introducing more goups with diffeent, age , form diferent clima, etc. Of course it is up to authors to decide  how to improve selection but probably from the large number of investigated patients a part of them have different tooth  characteristics  

c) the study used two type of methods but the spectrometry nowadays  introduced mapping 

d) conclusion has to be based on more different results and newer papers

Comments on the Quality of English Language

English language to my best knowledge is suitable for a scientific journal 

Author Response

Tooth colour is an important  factor in the fabrication of dental prostheses, despite the fact that may be is not  a determining one as afirmed  the paper abstract. But of course is an interesting subject for both type of target groups uch as patients and  people working in the field. This is a paper merit, The goal  of the present study is evaluation   of the two methods of color measurement in dentistry, using dental guides and spectrophotometry respectively. Such  proposed research design  is also a good idea but the way to support and to performed the investigation need to be more rich, Before publication the paper has to be completed according to the following

-  Question: introduction is based on the old references the majority of titles being older than 15 years.  

- Answer : new concepts with newer bibliographical references from lines 69-84 have been added to the introduction. Thank you for your comment.

 -  Question: methods have to be performed in a modified research design. Selection of patients it  will be better to be not larger but more complex, introducing more goups with diffeent, age , form diferent clima, etc. Of course it is up to authors to decide  how to improve selection but probably from the large number of investigated patients a part of them have different tooth  characteristics  

- Answer: New descriptive sample data have been added on lines 98-100. Thank you for your comments

- Question: the study used two type of methods but the spectrometry nowadays  introduced mapping 

- Answer: In this case, the measurement method has been described in the material and method section in lines 113-131, to be able to compare the tooth measurement with mapping between human and spectrophotometer would be more complex, as in some mappings up to 10 tooth colours can appear that the human eye is not able to differentiate. However, we have also added comments and references to this in the discussion section.

- Question: conclusion has to be based on more different results and newer papers

- Answer: we have added current articles incorporating intraoral scanners and mapping to the discussion section. Thank you for your appreciation.

Thank you

Reviewer 4 Report

Comments and Suggestions for Authors

The paper “Clinical Comparative Study of Shade Measurement Using Two Methods: Dental Guides and Spectrophotometry” evaluated the concordance of the two methods of color measurement in dentistry, using dental guides (Vita Classical and Vita Toothguide 3D-Master, Vita Zahnfabrik) and spectrophotometry (Vita Easysahde Compact, Vita-Zahnfabrik) in relation to the lightness or value of natural teeth. 

The article is interesting, nevertheless some major flaws should be addressed.

“illuminated with artificial light with white light, and natural light through 2 windows of 1 meter wide by 1.5 meters high. Measurements were made inside the cabinet with a Sekonic dual spot l-778 photometer (Sekonic CO, Japan) to locate the areas of approximately 5,500 degrees Kelvin. “

The cabinet should have been not influenced by windows ope, since the variety of the light could have influenced shade taking procedures.

How many minutes were used to take the shade with shade guides or with spectrophotometer?

Please relate the findings with clincal practice: e.g. color match in anterior restorations: the authors could use the following as a reference:

Manauta J, Salat A, Putignano A, Devoto W, Paolone G, Hardan LS. Stratification in anterior teeth using one dentine shade and a predefined thickness of enamel: a new concept in composite layering--Part I. Odontostomatol Trop. 2014;37(146):5-16.

A graph could help the readers in understanding differences between subjective and objective color selection methods

Author Response

- Question: “illuminated with artificial light with white light, and natural light through 2 windows of 1 meter wide by 1.5 meters high. Measurements were made inside the cabinet with a Sekonic dual spot l-778 photometer (Sekonic CO, Japan) to locate the areas of approximately 5,500 degrees Kelvin. “

The cabinet should have been not influenced by windows ope, since the variety of the light could have influenced shade taking procedures.

Answer: The colour was taken in areas of the room where there were no shadows, as it is very difficult to take the tooth colour. However, it is important to illuminate the room with white artificial light and natural light so that the selection conditions were correct, and a photometer was used to measure the correct light points in the room. The suggestion you have made in lines 127-130 has been added to the text in the measuring process section.

- Question: How many minutes were used to take the shade with shade guides or with spectrophotometer?

- Answer: In the shade selection shot with dental guides, tooth-by-tooth measurements were taken. The spectrophotometer was only one shot with the 1 second machine. For the measurement with dental guides, 10-15 seconds per tooth was used, because if the gaze is fixed for more than 30 seconds, a phenomenon of accommodation of vision occurs, which can influence tooth colour acquisition.

- Question: Please relate the findings with clincal practice: e.g. color match in anterior restorations: the authors could use the following as a reference:

Manauta J, Salat A, Putignano A, Devoto W, Paolone G, Hardan LS. Stratification in anterior teeth using one dentine shade and a predefined thickness of enamel: a new concept in composite layering--Part I. Odontostomatol Trop. 2014;37(146):5-16.

- Answer: The article reference has been added in the discussion section.

- Question: A graph could help the readers in understanding differences between subjective and objective color selection methods

-Answer: Two graphs have been added to visually clarify the results obtained. Thank you for your feedback.

Thank you

Round 2

Reviewer 1 Report

Comments and Suggestions for Authors

all comments were added

Reviewer 2 Report

Comments and Suggestions for Authors

The authors addressed the comments. I recommend to accept the manuscript. 

Reviewer 3 Report

Comments and Suggestions for Authors

I do agree that the article after the  second revision is a much better  paper. The introduction was improved  enlarging the presentation of subject status of the art based especially on new references  from recent  bibliography.              The  research design was enriched with new methods. Discussion and Conlusion chapters are more consistent in the present form of the article

Comments on the Quality of English Language

English Language is suitable for for   a scientific paper and only moderate editing required

Reviewer 4 Report

Comments and Suggestions for Authors

The authors have provided all the requested improvements.